

# The external and data loose coupling for the integration of software units: a systematic mapping study

Juan Antonio Ruiz Ceniceros[1], José Alfonso Aguilar-Calderón[2], Roberto Espinosa[3] and Carolina Tripp-Barba[4]

[1] Posgrado en Ciencias de la Información, Universidad Autónoma de Sinaloa, Culiacán, Sinaloa, Mexico
[2] Cuerpo Académico Tecnología Educativa I+D+i, Facultad de Informática Mazatlán, Universidad Autónoma de Sinaloa, Mazatlán, Sinaloa, Mexico
[3] Departamento de Ingeniería en Computación e Informática, Facultad de Ingeniería, Universidad de Tarapacá, Arica, Chile
[4] Facultad de Informática Mazatlán, Universidad Autónoma de Sinaloa, Mazatlán, Sinaloa, Mexico

## ABSTRACT

Integration of legacy and third-party software systems is almost mandatory for enterprises. This fact is based mainly on exchanging information with other entities (banks, suppliers, customers, partners, *etc.*). That is why it is necessary to guarantee the integrity of the data and keep these integration's up-to-date due to the different global business changes is facing today to reduce the risk in transactions and avoid losing information. This article presents a Systematic Mapping Study (SMS) about integrating software units at the component level. Systematic mapping is a methodology that has been widely used in medical research and has recently begun to be used in Software Engineering to classify and structure the research results that have been published to know the advances in a topic and identify research gaps. This work aims to organize the existing evidence in the current scientific literature on integrating software units for external and data loose coupling. This information can establish lines of research and work that must be addressed to improve the integration of low-level systems.

## INTRODUCTION

Enterprises are typically made up of hundreds of home-made (in-house development) applications, purchased from third parties, legacy systems, or a combination of all of them, operating in multiple layers on different operating systems. Currently, the integration of systems acquired from third parties and legacies has become a major concern in companies. As a result, most of the applications used in the enterprise are heterogeneous, autonomous, and operate in a distributed environment. Heterogeneity has been considered one of the most severe problems to solve because it tends to cause interoperability problems. In particular, semantic conflicts, which occur when applications use different meanings for the same information item. The challenges are integration is not an easy job; the real challenges are made up of several business and technical issues (*Hohpe & Woolf, 2004*).

Corresponding author
José Alfonso Aguilar-Calderón,
ja.aguilar@uas.edu.mx

In the context of SE, this area is known as Enterprise Application Integration (EAI) (*Irani, Themistocleous & Love, 2003*). EAI deals with integrating a heterogeneous set of applications and systems in any organization to integrate them and communicate information between the systems facilitating interoperability. According to this, enterprise integration is achieved using different sets of integration tools, technologies, and methodologies to ensure that transformation, translation, and communication of information items are accomplished efficiently. So, advances in integration technology, mainly concerning middleware, provide new ways to design more agile and responsive business architectures.

The integration of systems acquired by third parties is a real problem, mainly due to the lack of information exchange between entities such as banks, suppliers, customers, among others. Continual changes in the information systems environment have become the most important challenge in enterprises. The applications to be integrated are usually developed by different teams that often do not focus on the integration as a relevant issue for them. Aiming at eliminating the integration challenges, EAI is proposed as a solution. Faced with this situation, the need arises for new architectures for EAI, particularly, those with which to improve loose coupling in integrating software units.

There are hardly any research work with regard to EAI in scientific literature such as *Soomro & Awan (2012)* and *Gorkhali & Xu (2016)*. *Soomro & Awan (2012)* reviewed industrial challenges and problems in a general form only with which the limitations and research lines are not deep enough. In addition, in the field of the EAI the technological platforms evolve rapidly as a consequence since the year of the publication of this article is more than 9 years it is required to update the research to the current time. Under other conditions, *Gorkhali & Xu (2016)* performed a systematic literature review focused on categorizing EAI on the basis of industries. Regrettably, the primary studies were limited to those one published in Science Citation Index (SCI) and Social Science Citation Index (SSCI) database. Therefore, this study does not include publications in other recognized sources as Elsevier's, PeerJ, IEEE or Scopus. There is a SMS described by *Banaeianjahromi & Smolander (2014)* to survey and analyse the available literature on determining the role of enterprise architecture in enterprise integration and also to identify gaps and state-of-the-art in research. This is a very limited area study since search exclusively for methodologies and trends change over time. Another research studies focused on a particular technique to process integration such as the proposal from *Cerqueira et al. (2016)* which presents a SMS that investigated the use of ontologies to deal with semantics in integration at process layer level. In this regard, the work from *Fusco & Aversano (2020)* is about an ontology-based approach for semantic integration of heterogeneous data sources named DIF (Data Integration Framework); this work is not particularly of the field of EAI itself. These papers allow us to go deeper into related topics but specifically none of them has addressed the specific issue regarding to resolve the external and data loose coupling for the integration at software units level. Therefore, a review is needed since it is conducted based on a scientific search strategy such as SMS. To the best of our knowledge, no systematic mapping study has been conducted in this particular topic.

The goal of this study is to provide the state-of-the-art as well as a map of existing literature in this area. Furthermore, its evolution over time is shown to enable improvement of the practice with the known research results and to identify gaps for future research. For this purpose, this review aims to present a comprehensive summary of the studies in this field during the span of 2008–2021. The contributions of this review are:

- This SMS includes quality literature from pre-defined resources and based on pre-defined inclusion/exclusion criteria. Therefore, out of the 3,178 full-text articles studied, 39 articles were included.
- The proposals found in the primary studies for loose coupling software unit integration are based in an environment conformed by Service-Oriented Architecture (SOA), Web Services, and Microservices. Usually there are implemented by a pre-defined data types structure at design time, leaving an immovable structure at runtime.
- A comprehensive discussion on the existing proposals and the research gaps in this area. Furthermore, some suggestions for new research directions are suggested.

The intended audience of this research work refers but not limited to software engineers, information technology managers, business integration practitioners, researchers related to the area of EAI at software unit level as well as novice student researchers.

This article is organized as follows: in "Enterprise Application Integration Fundamentals" the relevant concepts to the context of this work are presented. "Survey Methodology" introduces the research and conduction protocol applied. In "Mapping Results", are presented the mapping results. "Discussion" introduces a discussion of the findings arising from research. Finally, "Conclusions" presents the final considerations and future work.

# ENTERPRISE APPLICATION INTEGRATION FUNDAMENTALS

This section introduces fundamental concepts necessary for a correct understanding of the rest of the article. The concepts addressed are Enterprise Application Integration, Middleware, Software Unit, External Coupling, Data Model, Loose Coupling, PUB/SUB Architecture, Federated Database System, Belief-Desire-Intention Architecture, Hub-and-Spoke, Apache Camel, Message-Oriented Middleware, Service Component Architecture, Grid Computing, Microservices REST, Service-Oriented Architecture and Model-Driven Architecture. These concepts are used thorough this SMS for the sake of understandability and completeness.

## Enterprise application integration

Enterprise Application Integration (EAI) is a discipline that dates to the beginnings of software engineering and is mainly responsible for software systems interacting with each other without any problem, understanding that this interaction refers mainly to the exchange of information between systems. Basically, it must allow software systems to be

able to share data and functions between them, allowing the connection between heterogeneous data sources and applications. All this must be achieved thanks to the implementation of a middleware between the two. There is no limitation between the type of software systems that must share information. These can be open source, in-house developments, or commercial license software systems. The problem in EAI lies mainly in the fact that originally the software systems were not designed to interact with each other or together, which implies a series of situations to be solved to achieve that communication. The scope of the EAI is located mainly in the integration of software systems in business-to-business environment (B2B) (*Wong (2009)*). Enterprise application integration can realize an effective combination of various independent systems, as data exchange and data sharing between all processes of an enterprise. Thus it is ensured for all units of a corporation to operate over a database system together with the suppliers and customers to improve enterprises' productivity and efficiency (*Zhigang & Huiping, 2009*).

## Middleware

A middleware is software located between an operating system and the applications that run on it. Middleware enables communication and data interchange in distributed software systems. The term middleware first appeared in a 1968 NATO (North Atlantic Alliance) conference report, which aimed to define the field of software engineering and included software design, production, and distribution. The goal of that report was the interconnectivity between software systems, particularly those considered older can be connected with the new software in organizations. Using middleware allows users to make requests such as submitting forms in a web browser or allowing a web server to return dynamic web pages based on a user's profile.

Examples of middleware can be found in database, application server and message-oriented middleware. Each of these programs generally provide messaging services and the different applications communicate through messaging frameworks. There are several messaging frameworks: Simple Object Access Protocol (SOAP), Web Services, Representational State Transfer (REST) and JavaScript Object Notation (JSON), those one is explained in this section for a better comprehension of this research. The decision of which one to use depends on the requirements of the enterprise: service to be used or type of information to be communicated.

The middleware can also be used for distributed processing with actions that occur in real time simplifying the development and maintenance of complex distributed software (*Astley, Sturman & Agha, 2001*).

## Software units

According to *Hong & Wen-yue (2010)*, a software unit is a modular component of a program with well-defined interfaces and dependencies that enable offering or requesting a set of services or functions. It can even be a piece that performs some task, a function, a method, a class, a library (library), an application, a component, among others (*Chen,*

*2009*). These often interact with data collections to save, update, delete, and present information.

The elements defining a software component has been widely discussed for more than 20 years (see *Broy et al. (1998)*). The most commonly adopted definition of a software component is that issued in *Szyperski, Gruntz & Murer (2002)*, where it was defined as a unit of composition with contractually specified interfaces and explicit context dependencies only. This can be deployed independently and is subject to the design by third parties.

### External coupling

An External Coupling (EC) occurs when a two or more Software Unit (SU) share an external enforced data format, interface or communication protocol. SU can be an artifact, platform, application or an API (Application Programming Interface) that uses technology based on an XSD (XML Schema Definition) or (JavaScript Object Notation). Usually, the structure using as communication protocol HTTP (Hypertext Transfer Protocol) or HTTPS (Hypertext Transfer Protocol Secure) to connect bi-directionally. This effect creates an External Coupling as a result of the structure (XSD, JSON). In addition, when this effect occurs at data level is called Data Coupling.

### Data model

According to *Brodie (1982)*, a database model is the logical structure that the database adopts, including the relationships and constraints that determine how data is stored and organized, and how data is accessed. Likewise, a database model also defines what type of operations can be performed with the data, that is, it also determines how it is manipulated, also providing the basis on which the query language is designed. It is composed of a collection of mathematically theory to assist one to express the static and dynamic properties of data-intensive applications. Those properties can be static such as objects with their attributes and the relationships between objects or sometimes called associations. They also have dynamic properties that occur between objects, these operations have their attributes and relationships that allow transactions to be carried out. In general, virtually all database models can be represented by a database diagram. The most common is the entity-relationship (ER) model.

### Loose coupling

According to *Orton & Weick (1990)*, loose coupling is the product of many years of effort by organization theorists to combine the contradictory concepts of connection and autonomy. In computing and systems design (*Kaye, 2003*), a loosely coupled system is one in which each of its components has or makes use of the definitions of other separate components. The coupling of classes, interfaces, data and services are sub-areas that are included. It promotes four types of autonomy, and these are reference autonomy, time autonomy, format autonomy, and platform autonomy. In this respect, loose coupling is the opposite of tight coupling.

Loose coupling is applied in the design and development of distributed systems through transaction, queues provided by message-oriented middleware and interoperability standards. Hence, it is even used as an architectural principle and design goal in SOA (*Pautasso & Wilde, 2009*).

## PUB/SUB architecture

Publish/subscribe messaging, or PUB/SUB messaging is an architectural design pattern that allows a framework to exchange messages between publishers and subscribers as a mean for disseminating information (also called events) through distributed systems on wide-area networks. Particularly, PUB/SUB asynchronous service-to-service communication used in serverless and microservices architectures. It involves the publisher and subscriber relying on a message broker that relays messages from the publisher to the subscribers (*Wadhwa et al., 2015*). According to *Baldoni, Contenti & Virgillito (2003)*, the participants to the communication are the publishers, they submit the information to the software distributed system, and as subscribers, that express their interest in specific types of information. In this kind of architectural pattern, either message published to a topic is rapidly received by all of the subscribers to the topic. Pub/sub messaging is used to enable event-driven architectures, or to modularize applications in to increase the software system performance, specially reliability and scalability software system quality attributes.

## Federated database system

The interoperability between different information systems is one of the most critical aspects in the daily operation of many organizations. This concern has been increased with the proliferation of different databases, with different data models, which run on different platforms. The federated databases systems answer this problem by allowing available information from different sources of information, which can be heterogeneous, distributed, and autonomous. The diversity of programming languages and queries, data models, methods of integration, among others, determine different architectures of the federated database, which vary from strongly coupled to loose coupled. Federated Database System is conformed of autonomous components participating in a federation to allow partial and controlled sharing of data. Such architecture differs based on levels of integration with the component database systems and the extent of services offered by the federation (*Muñoz & José, 2009*).

## Belief-desire-intention architecture

This architecture, abbreviated as BDI is a reasoning model based on mental constructs used by intelligent agents. It allows the modeling of agents behaviors in an intuitive manner that complements the human intellect. BDI is based on the human reasoning pattern, known as practical reasoning. First, decide what to achieve (deliberation) and then how to do it (reasoning). The agent using this model intends to show a legitimate reasoning to achieve his goals by using his beliefs about the environment (*Puica & Florea, 2013*).

## Hub-and-spoke

Hub-and-Spoke is an architecture applied as middleware, which uses a central message broker. In this architecture, communication is made between each application (spoke) and the central hub. The broker functionalities include routing and message transformation to the receiver spoke. Hub-and-Spoke additionally can routing based on content, using information from the message header or some elements of the message body. The hub from the message content can determine the receiver spokes, through rules (*An, Zhang & Zeng, 2015*).

## Apache camel

Apache Camel is an open source Java framework that aims to make software integration easy and accessible, it is used as middleware. Implement EAI business integration patterns using an API to configure routing and mediation rules. It was developed by the Apache Software Foundation and acts as a tool for rule-based data routing and processing. In addition, it has connectivity with a wide variety of transport protocols and supports DSL (Domain Specific Language) to facilitate its implementation by defining classes with the concepts of the domain. Its architect is divided into three modules, the first one is the integration and routing module, in which the processes and components are connected through messages based on criteria defined by the user, these can be defined in Java, Scala, XML or Groovy. The second is the process module is used to manage and mediate messages between endpoints. Business Integration Patterns are implemented in this module. Finally, the third one is the component module provides an interface to communicate with the external world through endpoints that are specified as URI (Uniform Resource Identifier) (*Gosewehr et al., 2018*).

## Message-oriented middleware

Message-oriented Middleware (MOM) is a concept that involves the passing of data between applications using a communication channel that carries autonomous units of information called messages. Basically, it is a software infrastructure that supports the sending and receiving of messages between the information systems of a company. In a MOM-based communication environment, messages are normally sent and received asynchronously. Through message-based communications, applications are abstractly decoupled; senders and receivers never know each other. Instead, they send and receive messages to and from the messaging system. To achieve this, it is necessary to process the messages in a controlled way in an environment with a client/server architecture. The processing is carried out by means of a program that works as an intermediary between the messages, which is designed to manage several messages from different clients and once forward them to the corresponding server program. The middleware builds a communications blanket that avoids developers from dealing with different operating systems and network protocols. The middleware creates a communications layer that isolates developers from the complexity of different operating systems and network protocols. This middleware is commonly used in scenarios where problems related to interoperability can occur if the network is constantly changing.

MOM is commonly used in IoT (Internet of Things) applications as centralized message brokers facilitate device-to-device communication. This performance is achieved because MOM has special capabilities such as dynamic scaling, secure communication, and facilitates its integration with other tools. Additionally, this architecture provides several features such as (i) asynchronous and synchronous messages transmission; (ii) the ability to convert the data format according to the data contained in the messages to be compatible with the application who will receive it; (iii) loose coupling among applications; (iv) parallel processing of messages; (v) management of message preference levels (*Albano et al., 2015*). According to *Yongguo et al. (2019)*, the advantages provided by MOM regarding asynchronous and multi-point interaction and loosely coupling among members is accepted as the most promising solution for communication-interaction between different systems.

## Service component architecture

Service Component Architecture (SCA) is a software technology designed to provide a model for applications that follow service-oriented architecture principles. The main concern of this architecture is to provide an open specification allowing multiple vendors to implement support for SCA in their development tools and runtimes. This is why it offers specific support for various component implementation and interface types such as Web Services Description Language (WSDL) interfaces and Java classes with corresponding interfaces (*Fiadeiro, Lopes & Bocchi, 2006*).

## Grid computing

Grid computing is a computer system that coordinates different computers with a hardware and software infrastructure to solve large-scale problems. Generally, a grid is responsible for performing several tasks within a network, however, it can also work in specialized applications. The term used to define grid computing originates from an analogy with the electric power grid: we can connect to the grid to obtain computing power without worrying about where it comes from. Just like we do when we plug in an electrical device (*Jacob et al., 2005*).

Grid computing is designed to solve problems that are too big for a supercomputer while maintaining the ability to process many small problems. Basically, it is based on virtualization between technologies, platforms and organizations, that is, a distributed computing infrastructure that is evolving in support of the application between organizations and the sharing of resources through the use of open standards. Within the grid computing hardware and software infrastructure there is a variety of resources, such as programming languages, either on a network or through the use of open standards with specific guidelines to achieve a common goal. Grid computing operations are divided into two:

1. Data Grid: This is a set of services that provides individuals or groups of users with the ability to access, modify and transfer large amounts of geographically distributed data for research purposes.

2. CPU Scavenging Grid: Is a technique that uses instruction cycles in computers to avoid wasted during the time the device waits for input from the user or other slower devices.

## Microservices REST

The microservice architecture emerged as a new paradigm for programming applications employing the composition of small services, each running its processes and communicating *via* lightweight mechanisms. The term microservices was first introduced in 2011 at an architectural workshop to describe the participants' common ideas in software architecture patterns; it is a fresh concept in software architecture, highlighting the design and development of deeply maintainable and scalable software. Microservices manage growing complexity by functionally decomposing large systems into a set of independent services (*Dragoni et al., 2017b*). A microservice is a small, single service offered by a company. It derives from the distributed computing architecture that connects many small services rather than having one large service. The Microservice can then be delivered through a Representational State Transfer (REST) API. REST is a software architectural style that defines the set of rules to be used for creating web services. It allows requesting systems to access and manipulate web resources by using a uniform and predefined set of rules. Interaction in REST-based systems happens through the Internet's HTTP (*Webber, Parastatidis & Robinson, 2010*).

## Service-oriented architecture

The formal definition of the term Service-Oriented Architecture (SOA) was given by the SOA Working Group which is member of The Open Group. It remarks that SOA is an architectural style that created for a special form of thinking in terms of services and service-based development and the outcomes of services called service orientation. In this regard, an architectural style is a set of design decisions that can be applied to a recurring design problem and that can be parameterized for different contexts where that design problem appears and a service is considered a logical representation of a repetitive operation from the business logic, *e.g.*, check order, review expired products. One of the main characteristics of a service is that is self-contained but can be constituted by several services. The SOA architectural style has a set of special features that must be applied such as (i) it is based on the commercial activities of the company, those used in the real world, that is, it reflects the business processes of the company with the client or with other companies, (ii) services are named and represented as business processes or company rules that represent a description of the company. Services are implemented through service orchestration, and (iii) it must be implemented through standards that allow maintaining interoperability of services (*The Open Group, 2009*).

## Model-driven architecture

Model-Driven Architecture (MDA) (https://www.omg.org/mda/) standard is an approach to software design, development and implementation supported by the OMG (Object

Management Group). MDA provides guidelines for structuring software specifications that are expressed as models.

MDA is a three-layer architecture where in the first one, the Computational Independent Model (CIM) specifies the project requirements and through a series of model-to-model transformations (M2M) the models of the second level of the three-layer architecture are obtained. These are the Platform Independent Models (PIM), which lack specifications on the implementation technology, normally are represented using class diagrams. Finally, the PIMs become Platform Specific Models (PSM), which are obtained through M2M transformations. PSM models are converted to source code as specified in the tool that implements it. The difference among PIM and PSM models is that PSM models are represented using the platform implementation technology, *e.g.*, the programming language and data-base management system to use (*Aguilar et al., 2010*).

## SURVEY METHODOLOGY

To know different proposals to improve the loose coupling in the integration of software units around this topic, a Systematic Mapping Study (SMS) was conducted. SMS is a methodology widely used in research in the medical area. Recently, it has begun to be applied to the field of SE to classify and structure the research results published. To learn about advances in a topic and identify gaps in research, there is also the methodology known as Systematic Literature Review (SLR). It seeks to identify best practices (based on empirical evidence) by conducting an in-depth exploration of the studies, describing their methods and results.

The difference from SMS is that it seeks to provide a more general vision, and SLR is focused on gathering and synthesizing evidence (*Petersen et al., 2008*). The main goal of SMS is to provide an overview of the research area and identify the amount and type of research and the available results. It is also essential to map published frequencies over time to understand trends and identify forums where research in the area has been presented (*Kitchenham et al., 2010*).

In this work, the methodology for SLR described in *Petersen et al. (2008)* was used. However, the adaptation proposed in *Kitchenham et al. (2010)* is applied to adjust to the SMS.

The review procedure to follow is composed of five stages: (1) Definition of research questions, (2) Search for primary studies, (3) Selection of documents for inclusion and exclusion, (4) Classification schema, and (5) Data extraction and systematic mapping. Each stage is detailed next concerning how it was carried out for this research.

### Definition of research questions

To bring out the SMS, a total of three research questions (RQs) were designed. These questions allow us to categorize and summarize the current knowledge concerning *loose coupling* of software units within Enterprise Application Integration. The goal is to identify gaps in current research to suggest areas for further investigation and to provide useful knowledge for software architects practitioners. The RQs are described next.

RQ-1. Which are the main proposals to improve the loose coupling in the integration of software units? It is intended to know the technological contribution that the primary studies make in the challenge of software units integration. The contributions would be, *i.e.*, architecture proposal, framework, architectural pattern, a tool, *etc.*

RQ-2. Which technology architectures have been considered to improve loose coupling in software unit integrations? The question tried to analyze the technological architectures proposed until now to improve loose coupling in software unit integrations.

RQ-3. Which technology or frameworks have been developed for loose coupling in software unit integration? This question includes libraries, and its goal is to determine whether there is a lack of tools to assist developers in the loose coupling of software unit integrations.

The scope of the review was defined as recommended by *Kitchenham (2007)* as follows: Population: researchers, professionals, and entrepreneurs who should improve the integration of software units. Intervention: any study that contains the description of a software unit integration solution at the software unit (component) level. Study design: applications in industry or academic examples. Result: evolution over time of the use of software unit integration technologies at the component level.

## Search of primary studies

Primary studies were identified using search strings in scientific databases. The Springer, IEEE, CONRICYT[1], Google Scholar, arXiv, and DOAJ databases were used to select the primary studies in this work. It is important to point out that CONRICYT is a resource supported by the Mexican government that allows researchers to search for scientific articles in databases/editorials such as Web of Science and Elsevier, among others.

To search for the scientific production associated with the concept of *Loose Coupling* and *Enterprise Application Integration*, the keywords were defined to construct the search strings to be consulted. To do this, keywording was performed (*Petersen et al., 2008*). In the first place, the main concepts of the research were identified such as keywords. Then, similar terms (synonyms) or phrases that might also be used to describe these concepts were defined. Next, a thesaurus was consulted to find synonyms. After that, the search terms were combined using Boolean operators. It is essential to mention that a recommended manner to create a search string is structuring them in terms of population, intervention, comparison, and result (*Kitchenham, 2007*).

The selected search period is from the years 2008 to 2021. The restriction with respect to the time period is to achieve a focused approach, the search was narrowed down to published journal articles from 2008 to 2021. Moreover, since the field of EAI and especially at the data level integration, is relatively old and it has been during the last decade where these ideas have been most developed with the emergence of web services and cloud computing. Likewise, this period of time has been validated during the development of the systematic mapping, since all the studies in this area are located within that period of time. In this sense, generic search expressions were considered considering the structure of each research question, the essential terms, and identified synonyms.

[1] Consorcio Nacional de Recursos de Informacióon Científica y Tecnolóogica of Mexico.

**Table 1 Structuring search strings.**

| Most important terms | Synonyms, terms and, topics | Search expression |
|---|---|---|
| **RQ-1** | | |
| "Loose Coupling", "Enterprise Application Integration", "Software Integration Proposal", "Coupling", "Integration" | "EAI", "Software Unit Integration" | Loose Coupling OR Enterprise Application Integration OR Software Integration Proposal OR (Coupling AND Integration) |
| **RQ-2** | | |
| "Loose Coupling", "Enterprise Application Integration", "Software Integration Proposal", "Coupling", "Integration", "Technological architectures" | "EAI", "Software Unit Integration" | (Loose Coupling AND Technological architectures) OR (Enterprise Application Integration AND Technological architectures) OR (Software Integration Proposal AND Technological architectures) OR ((Coupling) AND (Integration) AND (Technological architectures)) |
| **RQ-3** | | |
| "Loose Coupling", "Enterprise Application Integration", "Software Integration Proposal", "Coupling", "Integration", "Framework" | "EAI", "Software Unit Integration", "Library" | (Loose Coupling AND Framework) OR (Enterprise Application Integration AND Framework) OR (Software Integration Proposal AND Framework) OR ((Coupling) AND (Integration) AND (Framework)) |

The expressions were constructed using logical operators for searches (AND and OR). Table 1 shows how the search string was generated concerning the research questions.

Based on the defined search strings detailed in Table 1, a bibliometric analysis was performed using Vosviewer (https://www.vosviewer.com) software. To identify the tendencies of the literature on *loose coupling in the integration of software units*, an initial analysis of co-occurrences of keywords was conducted based on articles with at least five occurrences. The research resulted in six clusters (see Fig. 1) involving eighty-one keywords. The clusters are (1) loose coupling, (2) enterprise application integration, (3) management, (4) performance, (5) model and (6) impact.

## Selection of documents for inclusion and exclusion

An initial step was completed to remove duplicates. Hence, it was structured in such a way that the name of the technology and all the references of the main study in which it was found were listed using a set of tags with which an initial verification of the information could be performed. In case of tags reporting unreliable information, the primary studies were revised again to solve inconsistencies.

Inclusion and exclusion criteria were established to determine the relevance of the selection process of primary studies.

The defined inclusion criteria consist of (i) the search terms appearing at the working title or abstract considering the publication date from 2008 to 2021, (ii) research articles written in English language, (iii) articles with the full text available in the bibliographic source, and (iv) articles with potential to answer some of the RQs from Section "Definition of Research Questions". Likewise, the abstract refers to the problem covered by the corresponding research question.

Concerning the exclusion criteria, it was decided to exclude publications written in non-English languages, panel discussions, presentation slides, and tutorials. All articles

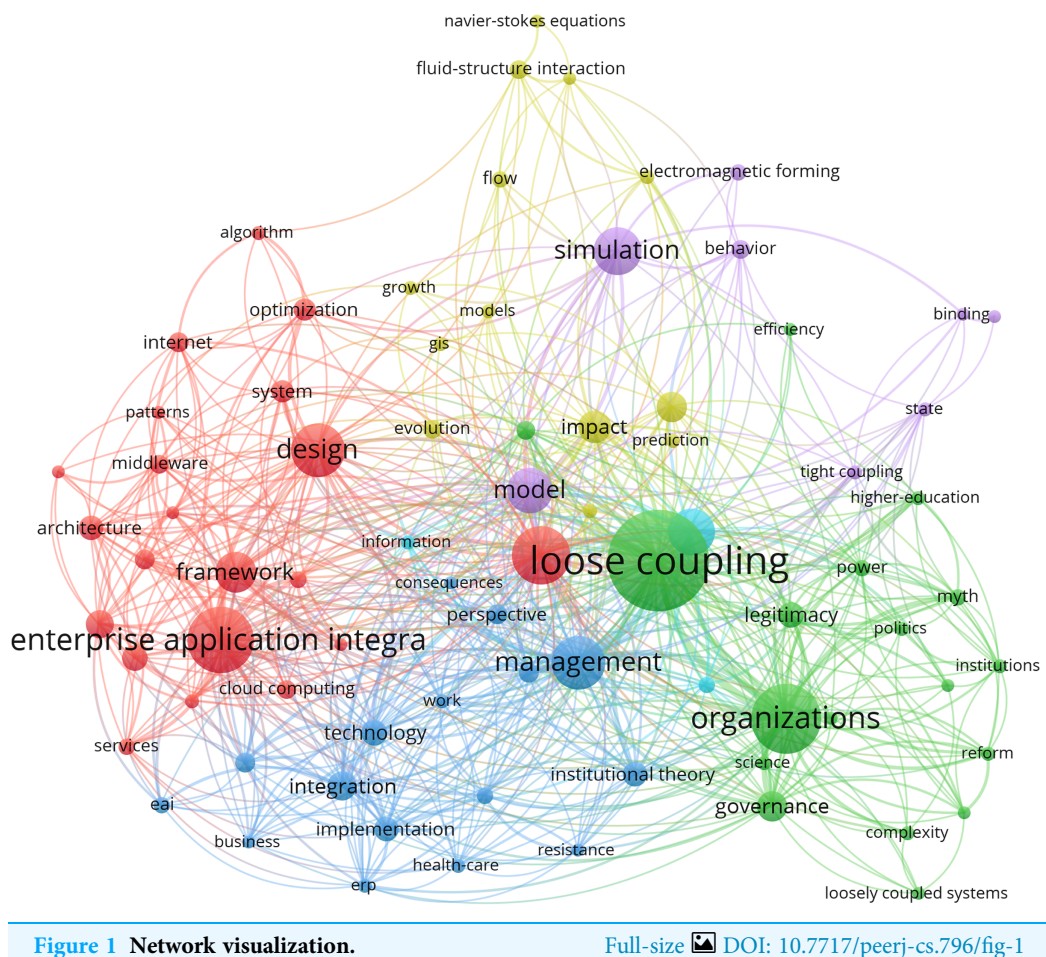

**Figure 1  Network visualization.**

retrieved unrelated to the topic explored were also excluded, as well as duplicate documents of the same study. Articles that highlight the initial draft of work in progress type articles were also excluded. To assess the relevance of each primary study to our topic, an iterative procedure was followed: all primary studies were analyzed on the basis of their title, abstract, and their full text.

The selection process consists of three stages conducted sequentially by three reviewers (two researchers and one collaborator). In the first stage, each reviewer applied the inclusion and exclusion criteria for the title, abstract, and keywords from the articles found. In the next stage, each reviewer applied the same criteria to a set of articles assigned to him, which now includes the introduction and conclusion. Afterwards, a set of candidate articles (see the second row of Table 2) were obtained. In the third stage, the candidate articles were analyzed. In this form, out of a total of 3,095 articles, a total of 39 primary studies were selected for mapping (see the third row of Table 2).

## Classification scheme

For the SMS, a systematic process was followed, which is shown in Fig. 2. It is a way to reduce the time necessary to develop the classification scheme and guarantee that it considers the existing studies. It was performed in two steps: (1) reading the summaries of

**Table 2 Search result and filtering divided by source.**

| Search engine | Springer | IEEE | CONRICYT | Schoolar Google | arXiv | DOAJ | Total |
|---|---|---|---|---|---|---|---|
| Search results | 593 | 889 | 319 | 900 | 293 | 184 | 3,178 |
| Candidates | 583 | 864 | 306 | 881 | 278 | 171 | 2,912 |
| Primary studies | 4 | 13 | 4 | 8 | 5 | 5 | 39 |

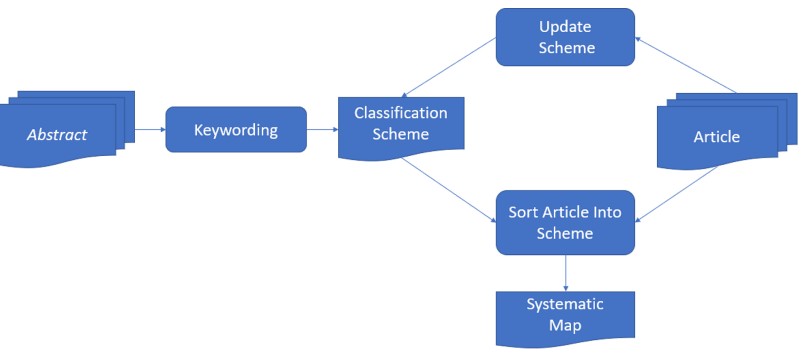

**Figure 2 Building the classification scheme.**

the articles found and (2) searching keywords and concepts that reflected the article's contribution to identifying the research context. It caused the combination of a set of different keywords of the research articles reviewed, allowing the development of a high-level understanding of the nature and contribution of the study, which allowed to find a group of representative categories of the underlying population. In relation to, if abstracts are of lower quality to allow choosing meaningful keywords, you may select to study the introduction or conclusion sections of the article. If a final set of keywords has been chosen, they can be grouped and used to form the map categories (*Irani, Themistocleous & Love, 2003*).

Businesses must be agile and flexible, and IT managers are being asked to deliver improved functionality while leveraging existing IT investment. Mostly of business organizations nowadays are using packaged software for their key business processes and goals (legacy business applications). Some of them are those who attend activities with respect to Enterprise Resource Planning (ERP), Supply Chain Management (SCM), Customer Relationship Management (CRM), and Electronic Commerce (EC). These systems assist business organizations in supporting their operational and financial goals. Bearing this considerations in mind, the classification scheme is divided in: *Proposals*, *Architectures* and *Technologies/Libraries*.

The first classification scheme (Proposals) refers to loose coupling software unit integration proposed to integrate these packaged software applications with each other regardless of its conformation. This scheme can include a framework, an architecture, a full computer package, or software abstractions that facilitate the development of loos coupling software unit integration, *i.e.* systems where elements can be easily added, removed, or replaced without needing widespread changes across the system. The second classification scheme refers to architectures that have been considered to improve loose coupling in

software unit integration. The goal is to analyze the technological architectures proposed at loose level components until now, as well as the emerging architectural styles for developing and integrating enterprise applications. The third classification scheme refers to technologies and libraries that have been developed for loose coupling in software unit integration. In this concern, the organizational and technical framework to enable an enterprise to deliver self-describing and platform independent business functionality is considered.

The classification scheme used can be consulted at the cloud (https://figshare.com/s/a65784c26931b96570fb), where it is possible to see the classification according to proposals, architectures and technologies.

### Data extraction and systematic mapping

Once defined the classification scheme in "Classification Scheme", the relevant articles were then classified in order to perform the data extraction. As shown in Fig. 2, the classification scheme works while extracting data, such as adding new categories or merging and splitting existing them. In this step, a spreadsheet is used to document the data extraction process which contains each category in the classification scheme (proposals, architectures and technologies). When data is entered into the schema, a brief explanation was provided, detailing why the article should be in a particular category (*Irani, Themistocleous & Love, 2003*).

## MAPPING RESULTS

This section presents and analyzes the results obtained after conducting the data extraction process from the primary studies. Several articles were found in the literature focused on different aspects of EAI. The selected studies provided relevant knowledge on the research questions. It is important to remark that the 39 primary studies found provide an answer to more than one research question. These are answered below:

RQ-1. Which are the main proposals to improve the loose coupling in the integration of software units?

The main proposals obtained from the publications were those based on SOA, Web Services and Microservices. These proposals implemented different forms to perform the loose coupling between the software units, using techniques based on asynchronous messages through middleware queues and topics. Some of them used to make this task standardized service contracts such as WSDL (Web Services Description Language) for Web Services and Microservices. This is done by the implementations of protocols such as SOAP (Simple Object Access Protocol), HTTP, REST with the help of the data schema standard such as XSD (XML Schema Definition). For XSD, the data travels in an XML format (eXtensible Markup Language); in the case of REST the format used is JSON. Another alternative that arises is through the construction and development of Frameworks, Constraints, and Models of Metadata. The use of Federated Database Systems is another way to establish loose coupling at the data level using the creation of Data Models where information is exchanged based on predefined schemes called Canonical Data Models (CDM). Most recently, SOA implements an orchestration of Web

**Table 3 Main proposals for the improvement of the loose coupling between software units.**

| RQ-1 Proposals based on | Quantity | References |
|---|---|---|
| SOA | 10 | *Green (2013), Voican (2012), Cuadrado et al. (2008), Herrera Quintero et al. (2010), Devi et al. (2014), Kim (2009), Hong & Wen-yue (2010), Chen (2009), Coronado-García et al. (2011), Deng et al. (2008)* |
| Web Services | 7 | *Risimic (2016), Beer & Hassan (2018), González & Ortiz (2013), Monfort & Hammoudi (2009), Huang & Zhang (2010), Ji (2009)* |
| Microservices | 5 | *Dragoni et al. (2017a, 2017b), Parizi (2018), Shadija, Rezai & Hill (2017), Gómez (2018)* |
| Middleware | 3 | *Antipov, Antipov & Pylkin, 2016, Cranefield & Ranathunga (2013), de los Ríos (2016)* |
| Models of Metadata | 2 | *García & Montoya (2011), Muñoz & José (2009)* |
| Library and Framework | 2 | *Weyns & Georgeff (2009), Ma, Tang & Wang (2009)* |

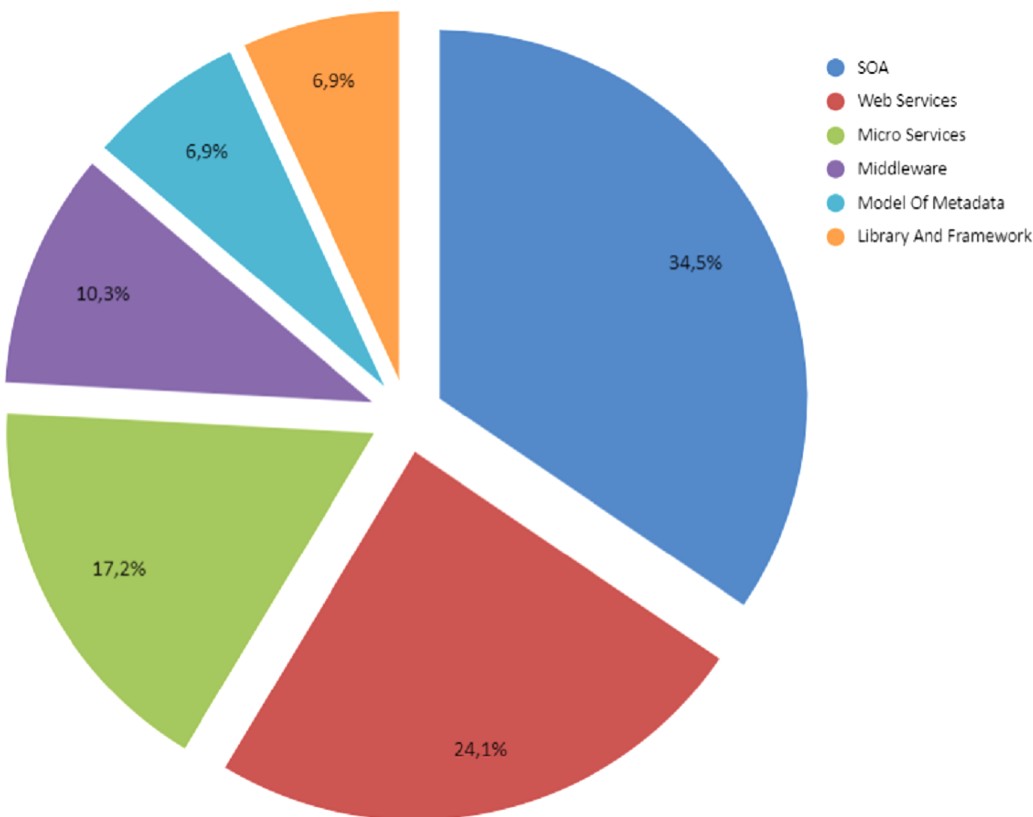

**Figure 3 Percentage of distribution for the main proposals found in the primary studies.**

Services and Microservices for loose coupling between the software units. Next, Table 3 summarizes the main proposals collected from primary studies.

From the proposals found in the publications, 34.5% were implemented with SOA, 24.1% through Web Services, and 17.2% using Microservices, representing approximately 76% of the proposals as shown in Fig. 3. The rest is composed of several technologies of

**Table 4  Main architectures used to improve loose coupling in software units integration.**

| RQ-2 Proposals | Quantity | References |
|---|---|---|
| SOA | 16 | *Voican (2012)*, *Cuadrado et al. (2008)*, *Herrera Quintero et al. (2010)*, *Devi et al. (2014)*, *Kim (2009)*, *Hong & Wen-yue (2010)*, *Chen (2009)*, *Coronado-García et al. (2011)*, *Risimic (2016)*, *Beer & Hassan (2018)*, *González & Ortiz (2013)*, *Monfort & Hammoudi (2009)*, *Martins et al. (2007)*, *Martnez & Pérez (2010)*, *Qu, Chen & Yang (2009)* |
| Microservices REST | 5 | *Dragoni et al. (2017a, 2017b)*, *Parizi (2018)*, *Shadija, Rezai & Hill, 2017*, *Gómez (2018)* |
| PUB/SUB | 2 | *Green (2013)*, *Antipov, Antipov & Pylkin (2016)* |
| Hub and Spoke | 1 | *Mohan et al. (2013)* |
| Camel Apache Architecture | 1 | *Cranefield & Ranathunga (2013)* |
| Reference Architecture | 1 | *de los Ríos (2016)* |
| Message Oriented Middleware | 1 | *Gutiérrez, Garca-Castro & Mihindukulasooriya (2013)* |
| Federated DataBase Architecture | 1 | *Muñoz & José (2009)* |
| BDI Architecture | 1 | *Weyns & Georgeff (2009)* |
| Intermediate Layer Architecture | 1 | *Lehsten, Gladisch & Tavangarian (2011)* |
| SCA Architecture | 1 | *Ma, Tang & Wang (2009)* |
| GRID Computing Architecture | 1 | *García & Montoya (2011)* |

which stand out the use if Middleware, Federated Databases Systems, and Frameworks (see Section "Enterprise Application Integration Fundamentals").

RQ-2. Which technology architectures have been considered to improve loose coupling in software unit integration's?

From the review made to the primary studies, it was found that a series of architectures were applied in the integration of software units to improve the loose coupling in the mentioned integration's. The architectures most implemented for this purpose were SOA and Microservices REST. SOA comes to be an orchestration of technologies supported by existing communication protocols such as SOAP and HTTP. Microservices based on REST is the second most used architecture for this purpose. Table 4 presents each one of these proposals found in the primary studies. As shown in Fig. 4, SOA architecture represents 50% of the implementations found in the primary studies, Microservices represents 15.6%, the 6.3% the PUB/SUB architecture; these, are as a whole 72%. However, there are other architectures that have been implemented such as PUB/SUB (*Green, 2013*; *Antipov, Antipov & Pylkin, 2016*), Hub & Spoke (*Krishna Mohan et al., 2013*), Camel Apache (*Cranefield & Ranathunga, 2013*), MOM (Message-Oriented Middleware) (*Gutiérrez, Garca-Castro & Mihindukulasooriya, 2013*), Federated Database Systems (*Muñoz & José, 2009*), BDI (Belief Desire Intention) (*Weyns & Georgeff, 2009*), Intermediate Layer (*Lehsten, Gladisch & Tavangarian, 2011*), SCA (Service Component Architecture) (*Ma, Tang & Wang, 2009*) and Grid Computing (*García & Montoya, 2011*), which is not widely used. This corresponds to the rest 28% of the total of primary studies.

In the case of the implemented architectures, Fig. 4 shows that the SOA architecture represents 50% of the implementations found in the primary studies, Microservices represents 17.6%, the 11.8% through Web Services; these are as a whole approximately

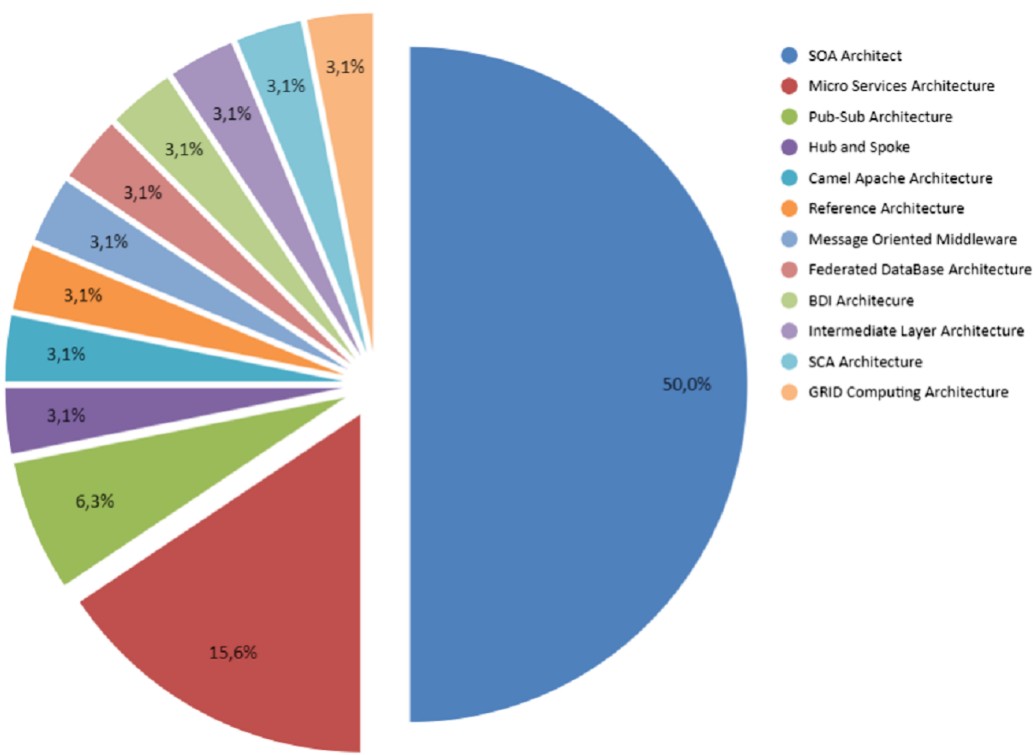

**Figure 4 Percentage by architecture found that has been implemented according to the primary studies obtained. From RQ–2.**

80%. The rest 20% a series of diverse architectures such as PUB/SUB, Camel Apache, Federated Database Systems (*Muñoz & José, 2009*), BDI (*Weyns & Georgeff, 2009*), Intermediate Layer and SCA (*Ma, Tang & Wang, 2009*).

RQ-3. Which technology or frameworks (including libraries) have been developed for loose coupling in software unit integration?

Several technologies were found in primary studies to address loose coupled in software unit integration, most of them being applied to architectures already validated such as those based on architectural patterns or archetypes. These are SOA, SOAP, WSDL, XSLT, ESB (Enterprise Service Bus), XML, XSD, BPEL (Business Process Execution Language), JMS (Java Message Service). For this reason, another set of technologies found for these purposes are the Microservices REST, and JSON. The primary studies also reported the application of Web Services with WSDL, SOAP, XML and XSD for integration's. Furthermore, technologies such as PUB/SUB, JMS, Queue, Topics, MDB (Message Driver Bean) were used to for software unit integration. Additionally, a set of technologies with fewer implementations with regard to software units integration were found, these are Apache Camel which is an open-source integration framework for data production or consumption, HL7 (Health Level Seven) (*García & Montoya, 2011*), Dublin Core, CORBA (Common Object Request Broker Architecture), RMI (Java Remote Method Invocation), Canonical Model, ODBC/JDBC (Open Database Connectivity/ Java Database Connectivity) (*Muñoz & José, 2009*), Self Adaptive, AI (Artificial Intelligence),

**Table 5 Main technologies used to improve loose coupling.**

| PI-3 Proposals | Quantity | Primary studies |
|---|---|---|
| SOA, SOAP, WSDL, XSLT, ESB, XML, XSD, BPEL, JMS | 17 | *Green (2013)*, *Voican (2012)*, *Cuadrado et al. (2008)*, *Herrera Quintero et al. (2010)*, *Devi et al. (2014)*, *Kim (2009)*, *Hong & Wen-yue (2010)*, *Chen (2009)*, *Coronado-García et al. (2011)*, *Deng et al. (2008)*, *Beer & Hassan (2018)*, *Monfort & Hammoudi (2009)*, *Qu, Chen & Yang (2009)*, *Sánchez, Aguilar & Exposito (2018)*, *Nazih & Alaa (2011)*, *Ruiz, Dueñas & Cuadrado (2008)* |
| Microservices REST, JSON | 6 | *Dragoni et al. (2017a, 2017b)*, *Parizi (2018)*, *Shadija, Rezai & Hill (2017)*, *Lendak et al. (2010)*, *Gómez (2018)* |
| Web Services, WSDL, SOAP, XML, XSD | 4 | *Risimic (2016)*, *González & Ortiz (2013)*, *Huang & Zhang (2010)*, *Ji (2009)* |
| PUB/SUB, JMS, Queue, MDB | 2 | *Antipov, Antipov & Pylkin, 2016*, *Patri et al. (2014)* |
| Apache Camel Tech | 1 | *Cranefield & Ranathunga (2013)* |
| HL7-Dublin Core Tech | 1 | *García & Montoya (2011)* |
| CORBA-CanonicalModel | 1 | *Muñoz & José (2009)* |
| Self-Adaptive Tech | 1 | *Weyns & Georgeff (2009)* |
| SCA Tech | 1 | *Ma, Tang & Wang (2009)* |

KQML (Knowledge Query Manipulation Language), Neuronal Network (*Weyns & Georgeff, 2009*) and Service Component Architecture (*Ma, Tang & Wang, 2009*). Next, Table 5 summarizes the technologies extracted from the primary studies according to technology/frameworks implemented for loose coupling in software units integration.

Relating to applied technologies, as shown in Fig. 5, the most implemented technologies are those based on SOA, Microservices REST, and Web Services. At glance, SOA represents 50%, in second place are Microservices with 17% and Web Services at last with 12% of technological implementations. The rest 21% of the technology/frameworks represent a diversity of scientific and technical knowledge implemented, which includes standards, frameworks, libraries, technological patterns and communication protocols.

Entirely, 79% of the 39 primary studies give a solution in the loose coupling in software units integration through these technologies. It is important to mention that, since these were selected from the primary studies reviewed, these technologies represent a large percentage of the results obtained in the review. Likewise, their application and publication in scientific articles highlight the fact that they are considered as the main technologies used today for the integration of loose coupling software units.

To provide a solution in the loose coupling in software integration's, the scientific community made a prominent effort to solve this problem from 2008 to 2011. Regrettably, this attempt began to decline in the period 2012 to 2017. Nevertheless, it gained momentum again in the year of 2018. It is possible to see that in these periods from 2008 to 2021; SOA, Web Services, and Microservices were the technologies primarily implemented for this purpose leaving aside technologies such as XML (eXtensible Markup Language). The analysis presented in Section "Mapping Results" shows a trend towards the investigation of approaches and architectures to increase connectivity since this is the most significant issue for EAI. The classification by year of publication concerning the different proposed research questions are shown in Fig. 6.

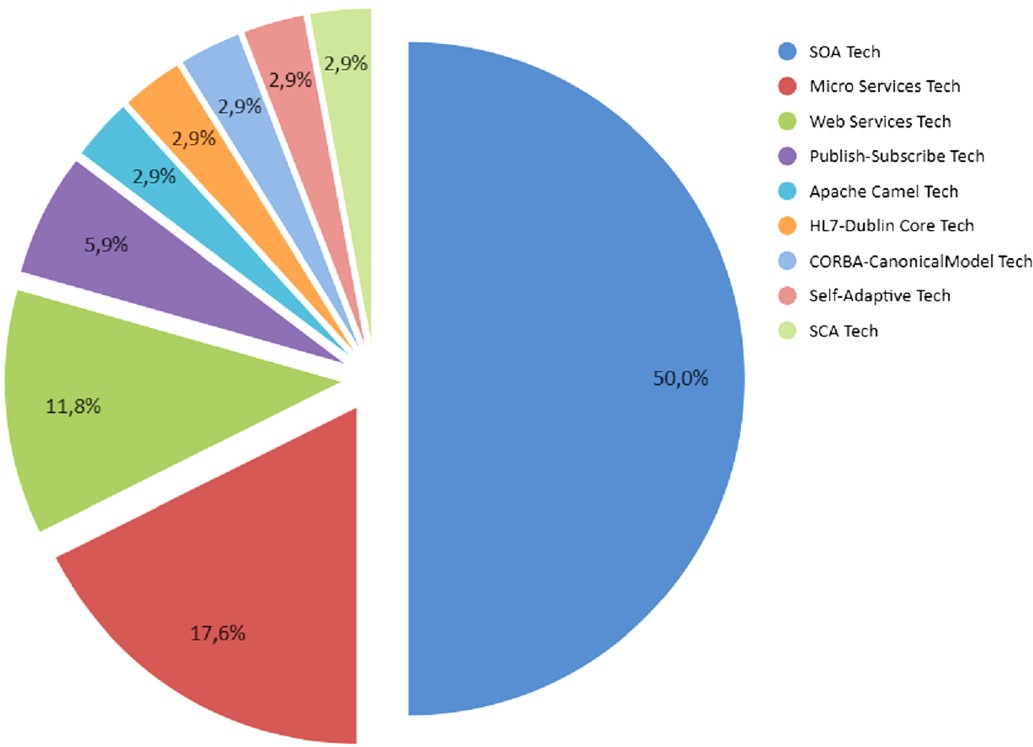

**Figure 5 Most implemented technologies. RQ-3.**

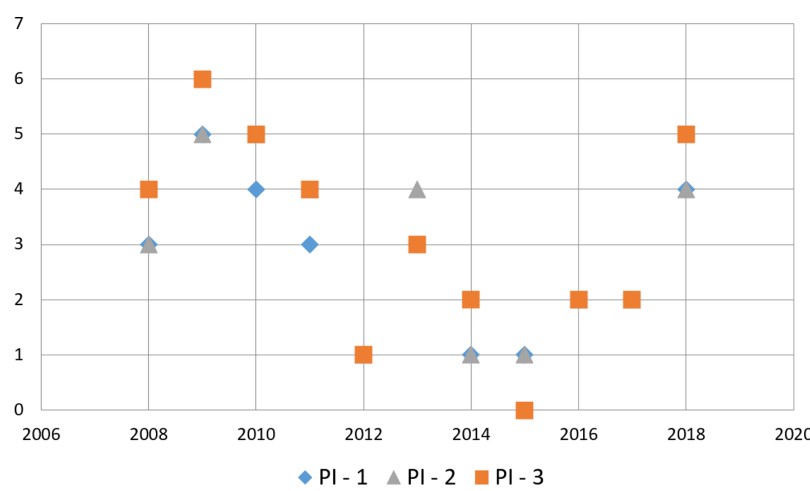

**Figure 6 Year of publication per research question.**

Most of the proposals found in the primary studies for loose coupling software unit integration are based in an environment conformed by SOA, Web Services, and Microservices. In this environment, the nodes of the network make their resources available to other participants in the network as independent services to which they have access in a standardized way, most of the definitions identify the use of Web Services using SOAP and WSDL in its implementation; however, it can be implemented using any service-based technology.

```
<?xml version="1.0"?>
<definitions name="StockQuote"

targetNamespace= "http://example.com/stockquote/definitions"
            xmlns:tns="http://example.com/stockquote/definitions"
            xmlns:xsd1="http://example.com/stockquote/schemas"
            xmlns:soap="http://schemas.xmlsoap.org/wsdl/soap"
            xmlns="http://schemas.xmlsoap.org/wsdl/"

        <import namespace="http://example.com/stockquote/schemas"
                location="http://example.com/stockquote/ stockquote.xsd"

        <message name="GetLastTradePriceInput">
            <part name="body" element="xsd1: TradePriceRequest"/>
        </message>

        <message name="GetLastTradePriceOutput">
            <part name="body" element="xsd1: TradePrice"/>
        </message>

        <portType name="StockQuotePortType">
            <operation name="GetLastTradePrice">
                <input message="tns:GetLastTradePriceInput"/>
                <output message="tns:GetLastTradePriceOutput"/>
            </operation>
        </portType>
</definitions>
```

**Figure 7 XML code fragment definition of <message> element in WSDL document for analysis purposes.**

To clarify the limitations found on these most implemented proposals, a WSDL structure for analysis is introduced in Fig. 7. This exemplifies how data types are defined at design time, leaving an immovable structure at runtime. In this concern, a WSDL document defines a set of services as collections of network endpoints or ports. The abstract definition of endpoints and messages is separated from their concrete network deployment or data format bindings. This separation enables the reuse of abstract definitions such as *messages* and *types of ports*. The *messages* are structured in such a way that they represent a description of the data that is exchanged that includes the ports as collections of operations to be performed. In that sense, the reusable link is made up of a protocol and contains the data format specifications for a particular type of port and a collection of ports defines a service.

The definition of network services in the structure of a WSDL document is made up of a series of elements, these are: Types, Message, Operation, Port Type, Binding, Port, and Service. These are detailed below:

1. *Types*: these are the data types of a system and are commonly defined in XSD structure.

2. *Message*: it is the structure that contains the data that is exchanged in a connection.

3. *Operation*: describe the actions supported by the service.

4. *Port Type*: are the operations supported by various endpoints.

5. *Binding*: this permits to specify the data format for a particular type of port.

6. *Port*, it is a combination of a link and a network address, it is a defined end point.

7. *Service*: is defined as a collection of related points.

```
<?xml version="1.0"?>
<schema targetNamespace="http://example.com/stock quote/schemas"
        xmlns="http://www.w3.org/2000/10/SMLSchema">

    <element name="TradePriceRequest">
        <complexType>
            <all>
                    <element name="tickerSymbol" type="string"/>
            </all>
        </complexType>
    </element>
    <element name="TradePrice">
        <complexType>
            <all>
                    <element name="price" type="float"/>
            </all>
        </complexType>
    </element>
</schema>
```

**Figure 8 Fragment XML code definition of element <xsd1: TradePrice> in WSDL document for analysis purposes.**

These elements are divided in two parts, the *Concrete* part, which defines the "how" and "where" and the *Abstract* for the definition of what the service does through the messages it sends and receives. Analyzing this structure, there is a deficiency in the high external and data level coupling in the integration of applications or software units because it is created before being used. According to Fig. 7, the element highlighted in yellow corresponds to the node (element) *<message>*, this element generates this coupling issue because it is created under a predefined fixed structure XSD before being used.

Analyzing the structure of the XSD showed in Figs. 7 and 8, if it is considered a change in the element highlighted *<TradePrice>*, this contract would impact all the applications or software units that are integrated into this structure. This which will cause a close coupling between them, consequently a high cost of maintenance and time. This is due to Fig. 7 exemplifies how data types are defined at design time, leaving an immovable structure at runtime. In Fig. 8, the label of the *<TradePrice>* element is observed, where the tags highlighted in yellow in the code represents the data and the properties of the structure called *<price>* and the type of this property *<float>* marking the strict way of receiving the exchange of information between software units. Therefore, if a change in the element *<TradePrice>* is considered, this contract would impact all the applications or software units that are integrated into this structure.

Clarifying this, let us assume this scenario, a Web Service or Microservice is consumed by a hundred clients where each one of them uses the same service contract (or WSDL document) to access to perform an operation. At once, if a change in the type of data is requested by some of those clients it will be necessary to create another new contract or WSDL document for each client that has changed. Furthermore, this scenario can be present if a client wants to integrate a new data type making it necessary to send one more data. This is important, since this entails having control and maintenance for each contract created, which implies that it is mandatory to create a hundred contracts in order to maintain the integration. This would represent a high cost in resources such as time, effort occasioning and increment in the project budget.

## DISCUSSION

This SMS has brought several perceptions into the research trends in EAI at the software unit level. These discernment's are discussed in this section.

Throughout the years, the main problem of EAI has been the communication and exchange of data between heterogeneous systems. In recent years, particularly in the period covered by this systematic mapping study, new technologies have emerged to develop software systems. Moreover, some existing technologies have been used in combination to provide robust frameworks for applications development that satisfy the needs of enterprises.

An issue that has been left behind over the years is the one related to considering an EAI project as an independent project reflecting its unique characteristics. No research was found on primary studies with regard to proposals for a step-by-step guide for EAI project definition and implementation. This is a critical issue to be studied since a software development project with a distributed architecture optimized for the data exchange is not the same as an EAI project. The SE guidelines allow obtaining a product that satisfies the customer's needs in a software development project, but this is not the case in EAI projects. There is no generic methodological approach for enterprises to implement it. This kind of project is considered one more component of a traditional software development process that implies a delay in its completion. Therefore, it is not considered an independent project that must be defined within a methodological approach only for EAI. In this sense, enterprises do not prioritize questions such as how to measure the value that an EAI project brings to them in the near future and how it could help significantly reduce maintenance costs. When they decide on an EAI project, they lack ad-hoc planning guided through a methodology for it.

Through the constant evolution of technological platforms for the development of software systems, many proposals for EAI have emerged in the scientific literature according to primary studies found. Most of them are based on SOA and microservices (see Fig. 4).

As shown in Fig. 6, the evolution of technology has been such that it has overcome the proposals that should exist to solve problems that the EAI has dragged on from the time when its early years. Furthermore, the scene is complicated again, as novel technologies appear to develop software systems, and as a result, new integration essentials are born with it. In this regard, research has dedicated much attention to succeeding simple data exchange among software units. Therefore, data privacy and security concerns have increased. Even though this area is out of the scope of this SMS, it is well-thought-out that research to date has overlooked the data security and privacy problems that emerged due to technological platforms and thus have not been adequately studied.

Over time the research in EAI does not provide a general framework for integration projects, nor at the level of data exchange, databases, or interfaces. In this regard, some efforts have been made, integration proposals have been industrialized, and scientific articles have even been published presenting solutions. The proposals help along with

being universal for EAI because they are based on MDA (see Section "Microservices REST"). The advantage proposed by MDA approaches to solve the EAI problem is that the same models created can be converted to the source code of the user's preference. This provides an advantage to EAI projects because by reusing the models, it is possible to generate the exact solution for different technological platforms (*Alahmari, De Roure & Zaluska, 2010*) and to build help along with new integrations in models to obtain the source-code for the integration. The advantage of this idea regards to improve a vital deficiency that is the re-configuration of business systems and not only the integration. Regardless of the importance of an approach like MDA, there has been not enough research in this field, as have many other topics at EAI.

The scientific literature emphasizes that current EAI solutions face a heterogeneity problem. Therefore, EAI solutions lack a robust and consistent integration approach supported by a methodology designed for that labor. Particularly, dedicated to the integration of heterogeneous enterprise applications that consider Requirements Engineering (RE) activities (*Aguilar et al., 2010*). Which can be modeled according to the necessities of the integration bearing in mind important software quality attributes such as security and privacy. Even that permits the requirements to be explicitly modeled on-demand as planned integration can be improved in execution. Nonetheless, it is predictable that the research will be uniformly concentrated through all activities of RE with emphasis within different research categories that provide supports to validated solutions such as experience reports, frameworks, and empirical studies that assist their decision-making. Nowadays, this is unaccomplished in EAI at the software unit level.

As mentioned above, enterprise applications are growing in number in different sectors of society. Almost all the business processes they handle have their software systems. These systems are based on different platforms and recent technologies. For this reason, they include multiple sources that lack interoperability, so dynamic EAI solutions are needed to solve integration problems. Dynamic EAI solutions can be achieved using SOA web services and with recent technology such as REST microservices. An advantage of these new technologies is the ease of integration at a low level that improves data exchange among applications. This enables interoperability through a controlled flow of data.

As new technologies evolve, it is important that EAI solutions adapt to these new technologies, but it is mandatory to increase the research in this area because the evolution is so fast that the problems that exist today will continue to exist tomorrow along with others more formed as a result of: (i) new implementation technologies, (ii) the growing demand of the telecommunications market, and (iii) a rapid change of the enterprise IT department as well as their technological environment. In addition, meanwhile, the main application of the EAI is the collaboration among systems of different enterprises (*e.g.*, banks for payments in sales, shipping services, as well as product providers). They dedicate a considerable amount of their budget more than human resources to maintain the exchange of data between the enterprises with which they interact in their business processes operating efficiently. Therefore, by creating interfaces at the software unit level that can integrate their applications, reducing the resources dedicated to maintenance will be possible. According to primary studies, SOA, as described by various authors as an

**Table 6 Primary studies selected for mapping (1/2).**

| Author(s) | Year | Title |
|---|---|---|
| Stephen Cranefield, Surangika Ranathunga | 2013 | Embedding agents in business applications using enterprise integration patterns |
| Reza M. Parizi | 2018 | Microservices as an Evolutionary Architecture of Component-Based Development: A Think-aloud Study |
| Nicola Dragoni, Schahram Dustdar, Stephan T. Larsen, Manuel Mazzara | 2017 | Microservices: Migration of a Mission Critical System |
| Nicola Dragoni, Saverio Giallorenzo, Alberto Lluch Lafuente, Manuel Mazzara Fabrizio Montesi, Ruslan Mustafin, Larisa Safina | 2017 | Microservices: yesterday, today, and tomorrow |
| Dharmendra Shadija, Mo Rezai, Richard Hill | 2017 | Towards an Understanding of Microservices |
| C. Punitha Devi, V. Prasanna Venkatesan, S. Diwahar, G. Shanmugasundaram | 2014 | A Model for Information Integration Using Service Oriented Architecture |
| Jongyeop Kim | 2009 | Mini-SOA/ESB Design Guidelines and Simulation for Wireless Sensor Networks |
| Danny Weyns, Michael Georgeff | 2010 | Self-Adaptation Using Multiagent Systems |
| António Martins, Pedro Carrilho, Miguel Mira da Silva, Carlos Alves | 2007 | Using a SOA Paradigm to Integrate with ERP Systems |
| K. K. Mohan, A. Verma, A. Srividya, G. Ravi Kumar | 2013 | A Practical Perspective on the Design and Implementation of Enterprise Integration Solution to improve QoS using SAP NetWeaver Platform |
| Stewart John Green | 2013 | An evaluation of four patterns of interaction for integrating disparate ESBs effectively and easily |
| Dejan Risimic | 2016 | An integration strategy for large enterprises |
| Vladimir Antipov, Oleg Antipov, Aleksander Pylkin | 2016 | Mobility support in publish/subscribe systems |
| Cristiana Voican | 2012 | Service Orientation in Distributed Automation and Control Service |
| Félix Cuadrado, Boni García, Juan C. Dueñas, Hugo A. Parada | 2008 | A Case Study on Software Evolution towards Service-Oriented Architecture |
| Elena Albertos Gómez | 2018 | Arquitecturas software para microservicios: una revisión sistemática de la literatura |
| Miguel Esteban-Gutierrez, Raúl García-Castro, Nandana Mihindukulasooriya | 2013 | A Coreference Service for Enterprise Application Integration using Linked Data |
| José Acosta Cano de los Ros | 2016 | Esquema de Referencia para Acoplamiento Débil entre Sistema Informático y Equipo de Producción |
| Edwin Montoya Múnera, Bernardo Augusto García Loaiza | 2011 | Integración de Repositorios Digitales en salud |
| Manuel Sánchez, Jose Aguilar, Ernesto Exposito | 2018 | Integrating SOA and MAS in Intelligent Environments |

architecture that can help solve not only the integration problem but also optimize integration techniques.

Finally, today, the EAI faces two significant challenges: syntactic and semantic integrations among enterprise applications at the level of software or data units. One reason for this is because each department/area built its software systems, making interoperability difficult. Primary studies show that it is essential to have a coherent semantic integration approach due to analysis. For this, the proposals continue to use a declarative definition for the data types and formats of the field that will facilitate the exchange of information in the integration configuration.

**Table 7 Primary studies selected for mapping (2/2).**

| Author(s) | Year | Title |
| --- | --- | --- |
| Ana Muñoz, José Aguilar | 2009 | Modelo Ontológico para la Integración de Bases de Datos Federadas |
| Luis Felipe Herrera-Quintero, Francisco Maciá-Pérez, Diego Marcos-Jorquera, Virgilio Gilart-Iglesias | 2010 | SOA-based Model for the IT Integration into the Intelligent Transportation Systems |
| Jose Luis Ruiz, Juan Carlos Dueñas, Felix Cuadrado | 2008 | A Service Component Deployment Architecture for e-Banking |
| Xiaogang Ji | 2009 | A Web-based Enterprise Application Integration solution |
| Luis Carlos Coronado-García, Jesús Alejandro González-Fuentes, Pedro Josué Hernández-Torres, Carlos Pérez-Leguízamo | 2011 | An Autonomous Decentralized Service Oriented Architecture for High Reliable Service Provision |
| Marina Nazih, Ghada Alaa | 2011 | Generic service patterns for web enabled public healthcare systems |
| José Vicente Berná Martínez, Francisco Maciá Pérez | 2010 | Model of integration and management for robotic functional components inspired by the human neuroregulatory system |
| Shiqi Ma, Jiangtao Tang, Dong Wang | 2009 | Process Based Application Level Architecture for RFID System |
| Mengjian Chen | 2009 | Research and Implementation on Enterprise Application Integration Platform |
| Imre Lendak, Ervin Varga, Aleksandar Erdeljan, Milan Gavrić | 2010 | RESTful web services and the Common Information Model (CIM) |
| Om P. Patri, Anand V. Panangadan, Vikrambhai S. Sorathia, Viktor K. Prasanna | 2014 | Semantic management of Enterprise Integration Patterns: A use case in Smart Grids |
| Wu Deng, Xinhua Yang, Huimin Zhao, Dan Lei, Hua Li | 2008 | Study on EAI Based on Web Services and SOA |
| Dongbing Huang, Wen Zhang | 2010 | Study on Enterprise Informationization Models |
| Hong Chen, Wen-yue Guo | 2010 | Study on enterprise Order Processing System based on SOA |
| Lili Qu, Yan Chen, Ming Yang | 2009 | The Coordination and Integration of Agile Supply Chain Based on Service-oriented Technology |
| Mohamed Ibrahim Beer, Mohd Fadzil Hassan | 2017 | Adaptive security architecture for protecting RESTful web services in enterprise computing environment |
| Laura GonzÁlez, Guadalupe Ortiz | 2013 | An ESB-Based Infrastructure for Event-Driven Context-Aware Web Services |
| Philipp Lehsten, Alexander Gladisch, Djamshid Tavangarian | 2011 | Context-Aware Integration of Smart Environments in Legacy Applications |
| Valérie Monfort, Slimane Hammoudi | 2009 | Towards Adaptable SOA: Model Driven Development, Context and Aspect |

## CONCLUSIONS

This work presents the results obtained after conducting an SMS. This research creates a synthesis of the current state of the art concerning loose coupling in software unit integrations in EAI. The goal is to provide scholars and practitioners with a comprehensive summary of recent research on this topic. Unfortunately, the literature lacks studies that report research work summarizing recent trends in this area.

For this, a total of 3,178 articles published in the literature were considered. These were extracted from scientific sources such as Springer, IEEE, CONRICYT, World Wide Web through Google Scholar, arXiv, and DOAJ. In the end, 39 primary studies (see Tables 6 and 7) were analyzed in-depth because they fulfilled the research questions proposed according to our inclusion and exclusion criteria detailed in Section "Selection of Documents for Inclusion and Exclusion".

SMS results shows that, in 13 years period, from 2008 to 2021, the same architectures or architectural patterns and technologies are still being used. Among them are SOA, Web Services, and in recent years Microservices. However, the technology used is limited because they continue using a WSDL data structure or service contract. Therefore, the coupling at the data level becomes a tight coupling. Additionally, this work remarks that EAI, particularly at loose coupling in software unit integration, has distinct requirements for e-commerce, banking, manufacturing, energy, and healthcare industries. Hence, to perform successful integration, various frameworks, architectures, and approaches are obligatorily needed. Thus, it is necessary to provide solutions within the architectures that offer us the goodness of decoupling the software units when they are integrated.

EAI at the software unit level is more than a technological trend. It is a form to consider structuring the information system to leverage the existing IT investments more effectively when developing new applications. Although the EAI has existed since the beginning of IS, it has constantly been evolving. As a result, integration has become more important than development in creating new applications for enterprises to deal with this problem.

The paper concludes that, as is well-known, EAI is a technology that helps an enterprise to achieve integration to inter and external software systems for data exchange. However, the integration technology solutions are often brand-named, which present interoperability issues because vendors restrict access to the code level, the complexity of services, and connectivity issues. Nevertheless, the distributed environment of enterprise applications outcomes in a complicated integration system. Consequently, new methodologies, platforms, protocols, technologies, and frameworks are still necessary to accomplish an all-inclusive EAI. In this regard, the future of EAI is based on platform-independent based solutions such as MDA, but lack of standards, robust frameworks for model-to-model (M2M) and model-to-text (M2T) transformations, problems related to tool support lacking usability with a poor user experience. This must be reviewed in terms of security and performance. Model-Driven technologies have been there since the 2000 year, and there are still the same issues. Likewise, changing dynamics of the application development process and usage pose a further challenge to achieve the desired result from EAI. Some of the current research suggests that much of EAI research is concentrated on developing a framework for EAI that can be used in different applications domains such as e-commerce, healthcare, and enterprise resource management systems. Additionally, the research on EAI is insufficient. There is an urgent demand to conduct more in-depth and significant research in developing new and enhanced frameworks and methodologies for EAI for cloud computing and IoT because these are technologies that are growing worldwide.

Some directions that future research from this SMS are suggested next:

1. The growing demand of the telecommunications market requires new implementation technologies, especially the development of robust frameworks considering cloud computing because enterprises applications drive their systems to this technology.

2. The emergence of new technologies, particularly technological platforms, demands more research at the EAI focused on the heterogeneity problem. The migration of old systems to new platforms such as cloud computing requires methodological guides and trained human resources who can direct the migration and integration with new technologies within the company.

3. The lack of research in the Requirements Engineering area within a methodological approach to implement an EAI project from the scratch. The idea to consider an EAI project as a different software development project must be adopted in enterprises because both have their particular characteristics. However, most of the time, they are considered the same or part of the main development software project because EAI's development cost is higher than a traditional approach. Furthermore, implementation takes more time and consumes more resources.

4. EAI is for data exchange among different technological platforms. Until now, microservices and SOAP are the most used technological platforms for that goal, according to Table 3. It is well known that microservices are distributed above several data centers, cloud providers, and host servers. Therefore, constructing an infrastructure through many cloud locations increases the probability of losing control and visibility of the application components. In this regard, data security and privacy issues must be of greater importance to researchers.

5. Consider to make more effort in Model-Driven based solutions. Considering MDA for framework development since their advantages are notorious and can be perfectly implemented in EAI ate software unit level. The solution for EAI can be generated in several programming languages just executing the M2M or M2T transformation over the same definition models.

As future work, we propose an architecture or pattern using a Dynamic Data Canonical Model (*Mork et al., 2014*) through the management of Agnostic Messages (*Celar, Mudnic & Seremet, 2017*). The messages will create a low external and data level coupling established in the service contracts that help integrate software units.

## ACKNOWLEDGEMENTS

The assistance provided by members of Cuerpo Académico Consolidado (Research Group) Tecnologa Educativa I+D+i (UAS-CA-303) from Universidad Autónoma de Sinaloa (Mazatlán, Mexico) was greatly appreciated.

### Funding

This work was partially funded by the Project UTA Mayor No. 8729-20 of the Universidad de Tarapaca, Arica, Chile.

## Grant Disclosures

The following grant information was disclosed by the authors:
Universidad de Tarapaca, Arica, Chile: 8729-20.

## Competing Interests

The authors declare that they have no competing interests.

## Author Contributions

- Juan Antonio Ruiz Ceniceros conceived and designed the experiments, performed the experiments, analyzed the data, authored or reviewed drafts of the paper, and approved the final draft.
- José Alfonso Aguilar-Calderón conceived and designed the experiments, performed the experiments, analyzed the data, performed the computation work, authored or reviewed drafts of the paper, and approved the final draft.
- Roberto Espinosa conceived and designed the experiments, performed the computation work, prepared figures and/or tables, authored or reviewed drafts of the paper, and approved the final draft.
- Carolina Tripp-Barba conceived and designed the experiments, analyzed the data, prepared figures and/or tables, authored or reviewed drafts of the paper, and approved the final draft.

## Data Availability

The data is available at Figshare: Aguilar-Calderón, Jose Alfonso (2021): Data Set for Systematic Mapping Study. figshare. Dataset. DOI 10.6084/m9.figshare.14888217.v2.

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
