# Peer review of "The external and data loose coupling for the integration of software units: a systematic mapping study"

_PeerJ Computer Science, doi:10.7717/peerj-cs.796_

## Round 0.1 · original submission · Major Revisions

The reviewers have shared their concerns, which I understand. I am quite confident that they can help you revise your paper so as to make it even better.

Reviewer 1 ·

Basic reporting

The paper presents a systematic mapping study (SMS) aimed at assessing the most recent (2008-2019) proposals, architectures, and technologies supporting the loosely coupled integration of software components, especially, in the field of Enterprise Application Integration (EAI).

The paper is clear, and well organised in structure, despite the English quality could be definitely improved but letting a native speaker read the paper: there are a number of recurrent typos.

A number of topics are surveyed in the paper, and for each of them, relevant entry point papers are referenced. However, the amount of information provided for each topic (e.g. in sec 2) is very concise, and barely sufficient for the reader which already has a basic understanding of the topic---therefore, I believe, more information should be provided for the readers which are not confident with those topics.

Images and tables are of adequate quality, despite not all images are vectorial. Figures 7 and 8 are somewhat low quality. Maybe a listing would more adequate.

I bet raw data could not be made available -- as it consists of the surveyed papers, I suppose -- yet, the authors share some intermediate data they have collected to produce the SMS, via a publicly available Google Sheet document. May I suggest sharing the data using a less volatile means? E.g. GitHub or Codeocean (maybe ask the editor what's the most adequate platform in their opinion).

Surveys of this sort are very important within the scopes of computer science and software engineering, and they cover a quite large amount of aspects of these disciplines. Furthermore, I'm not aware of any similar survey since the last decade, so I believe the contribution is useful and timely.

The introduction of the paper is clear, despite I struggle in understanding what does the "external" word -- which occurs into the title, abstract, and introduction -- refers to.

Experimental design

In my opinion, the study design could be improved in a couple of ways.

All the current research questions attempt to identify _which_ proposals/architectures/technologies are available into the literature to deal with loosely coupling and enterprise integration. For this reason, the results (sec. 4) are slightly more than an enumeration of keywords and references. This is for sure an extremely useful starting point, yet does not help the reader in understanding _how_ and to what extent the enumerated proposals/architectures/technologies favour loosely coupled integration. Maybe more queries and further analysis may be required to serve this purpose.

Also, I am not sure the current study design could be capable of capturing a shift in the meaning of the terms used to select primary studies. In other words, it seems to me that the authors are tailoring their queries to some terms which were extensively used one decade ago. Therefore, it is unsurprising that they are able to find proposals/architectures/technologies from around that period. It seems to me that the current design of the study does a good job in understanding the evolution of technologies of EAI in the last decade, despite it may struggle in figuring out whether new proposals/architectures/technologies have joined the game in the meanwhile.

Validity of the findings

At its current state, section 5 (discussion) is essentially useless. It spends a few pages discussing the rigidity of SOA approaches due to syntactical aspects of WDSL and... that's basically it. I believe a more elaborate discussion is needed.
An interesting discussion, in my view, should attempt to investigate the actual contribution of the selected proposals/architectures/technologies w.r.t. the problem of software integration (i.e. how they solve it), thus eliciting which problems can be currently considered solved and, possibly, which ones cannot.

Also, I believe the author should elaborate their conclusions (sec 6) better than this.
Currently, the section consists of 3 paragraphs: the first one summarises the methodology, the second one essentially states that the major issue of integration is WDSL (ok, but it sounds somewhat reductive to me), and the last one briefly summarises the importance of EAI, its reliance on proprietary technologies (which has never been discussed before in the paper) and some future works involving "Dynamic Data Canonical Model" and "Agnostic Messages"---both aspects are lacking references and have not been described before in the paper.

Additional comments

No additional comment.

Annotated reviews are not available for download in order to protect the identity of reviewers who chose to remain anonymous.

·

Basic reporting

The paper tackles an interesting problem related to how software units interconnect and focuses on the concept and current body of knowledge related to loose coupling. It performs a systematic mapping study to analyze the existing challenges in this domain.

The paper generally reads well, and the topic is very living to the journal. It is also essential for the software engineering community to be up to date with all the challenges exposed by software developers and researchers.

I have a few comments about the study design and results, which I will enumerate as I review the paper:

Experimental design

I did not understand why the authors have associated their research questions with the period of 2008 all the way to 2019.

Also, it was not clear to me why the authors have chosen to focus and this period exactly, keeping in mind that most of the references that the authors have used mainly to introduce all the concepts that they are surveying were actually pretty old and, to say the least, are before 2008. So, the choice of the period needs to be justified.

I think the authors should carefully explain how they performed the classification of the papers going from the selection of the potential venues to explore, all the way to the choice of documents using inclusion and exclusion criteria.

Yet, it was not clear to me how the authors have moved from a total of 3,095 articles all the way to 95 articles. If this has been done using the procedure outlined in figure number two, then this would be just along for manual analysis to what extent were able to do it without any buttons without missing any studies properly.

The authors have stated that the goal of the study is to identify gaps in current research to suggest areas for further investigation. However, the research questions outlined to fulfill this goal are most likely to be just exploratory and reporting numbers related to primary studies related to software unit integration and existing architectures and frameworks. Little is known about how exactly the Gap will be detected and how the authors have framed this Gap in terms of missing technology or with respect to the challenges related to software unit integration. In this context, I was expecting to see more limitations, errors, issues and problems related to each technology or framework and how existing studies are or not covering it.

Validity of the findings

This is also reflected in the results and mainly the discussion section that likes deep analysis of the limitations of existing papers. All the reflections made in the discussion are rather generic and known. Even the example brought by the authors about the limitation of defining data types during design time then dealing with them at the runtime is already a known problem. Besides being already known, this limitation does not seem to be extracted directly from the studies but rather it is a reflection of the authors of what they think it is a problem in these papers.

Section 4 which is called analysis of results seems to be just reporting results without any analysis. I did not see anything deeper than just specifying papers, which can be done using a classifier with proper training. The manual validation of the authors would have proposed a better understanding of these papers and what are the main challenges they are trying to solve. That would be better to improve the paper.

Additional comments

The authors mentioned the use of an excel sheet for their PS. It would be strongly recommended to share it with the community for replication and extension purposes.

---

## Round 0.2 · accepted · Accept

The reviewers have appreciated the revisions made, so—congrats :)

Reviewer 1 ·

Basic reporting

I am fairly satisfied with the improvements the authors performed in their last revision.
I believe my previous concerns have been reasonably addressed, and I have no further concerns.

Experimental design

Nothing new w.r.t. my previous review concerning the study design.

Validity of the findings

Nothing new w.r.t. my previous review concerning the validity of the findings.

Additional comments

I still believe that figures 7 and 8, as well as their descriptions in sec. 4, are maybe too much detailed and purpose-specific for the scope of this paper. I believe the authors may consider generalising that discussion to languages other than XSD or XML.

Also, there are still a few typos, e.g.:
- line 115, double ")"
- line 153, missing "JSON"

·

Basic reporting

NA

Experimental design

NA

Validity of the findings

NA

Additional comments

I would like to thank the authors for carefully addressing my comments. I no longer have any concerns with this paper.